# Evaluation of Sericin/Polyvinyl Alcohol Mixtures for Developing Porous and Stable Structures

**DOI:** 10.3390/biomimetics10010027

**Published:** 2025-01-05

**Authors:** Maria C. Arango, Leander Vásquez Vásquez, Akemy Carolina Homma Parra, Santiago Rueda-Mira, Natalia Jaramillo-Quiceno, Josep Pasqual Cerisuelo, Amparo Cháfer, Catalina Álvarez-López

**Affiliations:** 1Agroindustrial Research Group, Department of Chemical Engineering, Universidad Pontificia Bolivariana, Cq. 1 #70-01, Medellín 050031, Colombia; leander.vasquez@upb.edu.co (L.V.V.); akemy.homma@upb.edu.co (A.C.H.P.); santiago.ruedam@upb.edu.co (S.R.-M.); natalia.jaramilloq@upb.edu.co (N.J.-Q.); catalina.alvarezl@upb.edu.co (C.Á.-L.); 2Materials Technology and Sustainability (MATS), Department of Chemical Engineering, Universitat de València, Av. de la Universitat s/n, 46100 Burjassot, Spain

**Keywords:** silk sericin, polyvinyl alcohol, scaffolds, lyophilization, mixture

## Abstract

Fibrous by-products, including defective or double cocoons, are obtained during silk processing. These cocoons primarily contain fibroin and sericin (SS) proteins along with minor amounts of wax and mineral salts. In conventional textile processes, SS is removed in the production of smooth, lustrous silk threads, and is typically discarded. However, SS has garnered attention for its antioxidant, antibacterial, biocompatible, and anticancer properties as well as its excellent moisture absorption, making it a promising polymer for biomedical applications. Owing to its functional groups (carboxyl, amino, and hydroxyl), SS can blend and crosslink with other polymers, thereby improving the mechanical properties of sericin-based materials. This study explored the effects of different SS/polyvinyl alcohol (PVA) ratios on porous scaffolds fabricated via freeze-drying, focusing on the mechanical stability, water absorption, and protein release in phosphate-buffered saline (PBS). The scaffold morphology revealed reduced porosity with higher SS content, while increased PVA content led to material folding and layering. A greater PVA content enhanced water absorption, mechanical properties, and thermal stability, although SS release decreased. These results demonstrate that scaffold properties can be tailored by optimizing the SS/PVA ratio to suit specific biomedical applications.

## 1. Introduction

Natural polymers are extensively utilized in biomedical applications due to their inherent biocompatibility, biodegradability, and unique bioactive properties [1]. Among these, sericin (SS), a protein derived from silk cocoons, has attracted considerable attention [2]. During silk processing, sericin is typically removed to produce smooth, lustrous silk fibers, which are often discarded as waste [3]. However, the composition and functional groups of sericin, including hydroxyl, carboxyl, and amino groups, impart high chemical reactivity, which makes it suitable for the development of advanced materials. Sericin is non-immunogenic, promotes cellular attachment and proliferation, and accelerates wound healing by enhancing collagen synthesis and epithelialization [3]. Its antioxidant, antimicrobial, and anticoagulant properties highlight its potential biomedical applications [3,4].

In wound healing, sericin has been shown to retain skin moisture and improve tissue repair without triggering significant inflammatory responses [5,6,7,8]. For instance, sericin-based dressings have demonstrated the potential to accelerate wound healing and reduce the associated pain [4,9,10,11,12]. Despite these valuable properties, the inherent mechanical fragility of sericin limits its direct application in durable materials [13,14], necessitating its combination with other polymers to enhance its stability while retaining bioactivity.

To address this challenge, synthetic polymers, such as polyvinyl alcohol (PVA), are commonly incorporated [6]. PVA is a water-soluble, biodegradable polymer that is widely used in pharmaceutical and biomedical applications owing to its high mechanical strength, excellent film-forming ability, and flexibility [6,15]. Hybrid materials combining SS and PVA leverage the bioactive properties of sericin with the mechanical robustness of PVA, making them excellent candidates for biomedical uses, including wound dressings [16]. Stability ensures that the material maintains its structural integrity and performance during use, especially in moist environments such as wound beds [17]. Sericin/PVA systems offer a promising solution, but optimizing the ratio between these components is essential to balance bioactivity and mechanical properties.

Although previous research has highlighted the potential of SS/PVA blends and the incorporation of active agents [18], studies often lack a systematic evaluation of the effects of different SS/PVA ratios on the critical material properties. This knowledge gap limits the ability of these materials to be tailored to specific applications.

In this study, porous scaffolds composed of SS/PVA were fabricated via freeze-drying, and their morphology, mechanical stability, water absorption, and release of sericin in phosphate-buffered saline (PBS) were evaluated. By systematically analyzing the effects of varying SS/PVA ratios, this work aimed to identify the optimal formulation for applications in wound healing, where stability and structural integrity are crucial.

## 2. Materials and Methods

### 2.1. Materials

Silkworm cocoons supplied by the Corporation for the Development of Cauca Sericulture (CORSEDA) in Colombia were processed by cutting them into small pieces after removing dried pupae and impurities. PVA, a commercial product sourced from Sigma-Aldrich (St. Louis, MO, USA), was characterized by a hydrolysis degree exceeding 98–99% and a molecular weight of 146–186 kDa. Additionally, the biuret reagent, obtained from the laboratory of the Bolivarian Pontifical University (Medellín, Colombia), was used for protein quantification in solution. Phosphate-buffered saline (PBS), pH 7.4, 1X, was purchased from Gibco (Grand Island, NY, USA).

### 2.2. Procedure for Obtaining Sericin Powder

Silk sericin (SS) was obtained through a degumming process involving high temperature and pressure. Briefly, pre-cut silkworm cocoons were submerged in distilled water with a bath ratio of 1:30 (g/mL) and autoclaved at 120 °C for 30 min using an AV model autoclave (Phoenix Ltd., Araraquara, Brazil). The resulting SS solution was filtered through a fabric mesh to remove the residual cocoon fragments, particulates, and impurities. Subsequently, the SS solution was spray-dried using a spray dryer (BUCHI B-290 Labortechnik AG, Flawil, Switzerland) with an inlet temperature of 160 °C, airflow of 40 m^3^/h, and solution feed rate of 6.3 mL/min. The resulting SS powder was stored in a desiccator to preserve its properties until further use.

Our previous study [19] detailed the physicochemical characterization of this sericin powder, identified as a protein with a predominantly amorphous structure and low molecular weight fractions.

### 2.3. Fabrication of SS/PVA Scaffolds

The SS powder was dissolved in distilled water to prepare a 2% (*w*/*v*) sericin solution, following the methodology reported in other studies [20,21]. PVA powder was dissolved in distilled water to a final concentration of 2% (*w*/*v*) with constant stirring at 85 °C until it was fully dissolved. The PVA solution was then added to the sericin solution at the same temperature (85 °C), with continuous stirring for approximately 20 min to ensure proper polymer blending. The resulting solutions were prepared with varying SS/PVA ratios (75:25, 50:50, and 25:75), with 100% SS and 100% PVA serving as the controls. Each solution was poured into 2 mL wells on flat plates and frozen at −80 °C for 24 h. Following freezing, the samples were lyophilized (Labconco Corporation, Kansas, MO, USA) for 48 h to obtain the porous scaffolds. The scaffolds obtained were stored in a desiccator to preserve their properties until further characterization.

### 2.4. Characterization Methods

The characterization tests performed on the samples are described below.

#### 2.4.1. Fourier-Transform Infrared Spectroscopy (FTIR)

The chemical structures of the SS/PVA scaffolds at different ratios were analyzed using FTIR with an attenuated total reflectance module (ATR) on a Nicolet 6700 Series brand spectrometer (Thermo Electron Corporation, Beverly, MA, USA). A total of 64 scans were performed at a resolution of 4 cm^−1^ and a wavelength range of 4000–400 cm^−1^.

#### 2.4.2. Scanning Electron Microscopy (SEM)

The morphology of the SS/PVA hydrogels was analyzed using secondary electron HV-SEM (JSM-6490LV, JEOL, Tokyo, Japan) operating at 5 kV. Hydrogels were horizontally sectioned with a razor blade before being coated with an ~10 nm gold layer using Desk IV equipment (DENTONVACUUM, Moorestown, NJ, USA). Images were captured at 100× magnification and processed using ImageJ V1.53 software. In total, 100 pore widths were measured for each sample. Additionally, the chemical compositions of the cross-sections were assessed using SEM coupled with energy-dispersive X-ray spectroscopy (SEM/EDX). The acceleration voltage for capturing the images (100× and 50×) was 5 kV.

#### 2.4.3. Thermogravimetric Analysis (TGA)

The thermal behavior of the samples was analyzed using a thermogravimetric analyzer (TGA-Q500, TA Instruments, New Castle, DE, USA). Approximately 10 mg of each sample was tested under nitrogen atmosphere (50 mL/min) to prevent oxidative processes. The temperature ranged from 30 to 800 °C at a heating rate of 10 °C/min. The decomposition profiles and characteristic temperatures were determined by calculating the first derivative of the mass loss (DTG) from the TGA data.

#### 2.4.4. Water Absorption Capacity and Mass Loss

The water absorption capacity was evaluated following the method adapted from Mandal, Priya, and Kundu [22]. Prior to submerging the hydrogels in distilled water for a 24 h period, their dry weight was measured. To enable easy handling, the samples were positioned on a metal grate throughout the immersion process. After a specified period, the scaffolds were removed from the solution, and excess water was blotted using a cotton towel; their mass was measured immediately. The water absorption capacity was calculated using Equation (1).
(1)$$\mathrm{Water}\;\mathrm{absorption}\;\mathrm{capacity}\;\left(\%\right)=\frac{M_{w}-M_{0}}{M_{0}}\times 100$$
where *M_w_* (g) and *M*_0_ (g) are the masses of the samples in wet and dry initial states, respectively.

Following the water absorption test, the wet samples were dried at 60 °C for 24 h in a forced convection oven (BOV-V30F, Biobase, Jinan, China) and then weighed (*M_d_*) to assess the changes in mass. The mass loss was determined using Equation (2). All experiments were conducted in triplicate (*n* = 3).
(2)$$\mathrm{Mass}\;\mathrm{loss}\;\left(\%\right)=\frac{M_{0}-M_{d}}{M_{o}}\times 100$$

#### 2.4.5. Silk Sericin Release from the Scaffolds

The release of SS from the SS/PVA scaffolds at different ratios was evaluated by immersing the samples in 9 mL of 1X PBS solution (pH 7.4) at 37 °C under controlled conditions. After 24 h, 1 mL of the solution was collected for each test [23]. The SS content in the PBS solution was quantified using the biuret method, which detects proteins by reacting peptide bonds with a copper-containing reagent in an alkaline environment to form a purple complex [24]. A calibration curve was constructed using a standard protein solution at a concentration of 2% (*w*/*v*) [20]. The solutions were left to develop color for 30 min at room temperature before measurement using a UV–Vis spectrophotometer (DR 2700, Hach, Loveland, CO, USA). The blank consisted of 1 mL of PBS in 10 mL of biuret reactive, with absorbance set to 0.0 at 510 nm. All experiments were performed in triplicate.

#### 2.4.6. Compressive Strength

Compression tests were conducted on pristine SS, pristine PVA, and SS/PVA scaffolds (14.5 mm in diameter, 8.6 cm in height) using a universal testing machine (Instron 5582, Norwood, MA, USA) at a constant compression rate of 1 mm/min [25,26]. The test was terminated when 80% of the initial hydrogel thickness was reached. Each test was performed in triplicate, and the results are presented as mean ± standard deviation.

#### 2.4.7. Statistical Analysis

Differences in the process conditions affecting water absorption capacity, mass loss, PBS degradation, and protein release were evaluated using one-way ANOVA with multiple comparisons. Fisher’s LSD test was applied at a 5% confidence level to determine statistical significance.

## 3. Results and Discussion

### 3.1. Impact of Polymer Proportions on the Structure of SS/PVA Scaffolds

FTIR analysis was performed to identify the structural conformation of the scaffolds. The spectra of the pristine SS, PVA, and SS/PVA at different ratios are shown in Figure 1. In all samples containing SS, bands of amides from the protein were observed at the same vibration [27,28]. Amide I (1650 cm^−1^, blue arrow), amide II (1515 cm^−1^, red arrow), and amide III (1240 cm^−1^, black arrow) bands correspond to C=O stretching, N–H bending, and C–N stretching, respectively [7].

It is worth noting that the peaks of amide A and B (3000–3500 cm^−1^), with a higher intensity peak around 3200 cm^−1^ for the SS sample, were assigned to –OH stretching and the strong hydrogen bonding present in the β-sheet structures of the protein, as well as N–H stretching vibrations [29]. This peak progressively shifted in the 75/25, 50/50, and 25/75 SS/PVA scaffolds, indicating that blending promotes hydrogen bonding interactions [9,28].

The increased PVA content introduced additional hydroxyl groups, which interacted with the amide and hydroxyl groups of the protein through hydrogen bonding. This enhanced interaction resulted in a more pronounced shift, indicating a stronger network of physical interactions as the proportion of PVA increased in the blend.

In the PVA spectrum, a distinctive peak corresponding to alkyl vibrations was observed at 840 cm^−1^. This peak was also present in the SS/PVA samples, but decreased in intensity with a lower PVA content [9,28]. Additionally, the peaks at 2945 cm^−1^ and 840 cm^−1^ in the SS/PVA mixtures were attributed to PVA, with the former associated with methyl group vibrations and the latter with alkyl group rocking motions [30,31]. The FTIR results align with the previous literature [32,33,34], confirming the coexistence of SS and PVA without any evidence of chemical reactions between the components. Instead, the interaction between SS and PVA is governed by physical forces, such as hydrogen bonding, which contribute to the stability of the blend. The absence of new peaks or significant shifts in the spectra suggests that the structural integrity of both components was preserved, supporting the formation of a physically interconnected network [31].

Thermogravimetric analysis (TGA) and a derivative (DTG) of the pristine SS, PVA, and SS/PVA scaffolds are presented in Figure 2. All samples exhibited an initial decomposition stage below 210 °C, corresponding to a weight loss of 3–6%, which was attributed to the evaporation of free and bound water [20,35]. The second decomposition stage, occurring between 220 and 470 °C, is associated with the elimination of volatile compounds, degradation of amino acid residue side chains in proteins, and decomposition of PVA functional groups [36,37].

The PVA curve exhibited a maximum decomposition temperature of 257 °C, which was attributed to the breakdown of the side chains of this polymer [37]. Subsequently, a small peak was observed at 450 °C, which corresponds to the decomposition of the PVA main chain [36,37]. In contrast, the SS and SS/PVA mixtures exhibited similar behaviors, with maximum decomposition temperatures higher than that of the pure PVA. These peaks were observed at 302, 319, 326, and 324 °C for the SS, SS/PVA 75/25, 50/50, and 25/75 samples, respectively. Notably, as the sericin content increased, the thermal stabilities of the samples improved. This peak is associated with the decomposition of amino acid side chain groups and the cleavage of peptide bonds [38,39].

The peak observed at 450 °C in the PVA sample was also present as a small shoulder in the SS/PVA mixtures but diminished as the PVA concentration decreased. The SS-containing samples also exhibited a small shoulder at ~250 °C, corresponding to the decomposition of lower molecular weight SS [38]. These findings align with the literature, which reports PVA decomposition within the range 210–540 °C and sericin decomposition starting at 220 °C [39,40].

The SS/PVA samples exhibited higher decomposition temperatures than the pure SS and PVA control samples, demonstrating improved thermal stability. This enhancement is primarily attributed to the strong physical interactions between the SS and PVA chains, which restrict polymer chain mobility and require more thermal energy for decomposition, thereby increasing thermal stability. These findings align with the FTIR analysis, which highlights the physical crosslinking of the polymers in the blend and the strong intermolecular affinity between SS and PVA.

### 3.2. Morphological Analysis of SS/PVA Scaffolds

The morphologies of the cross-section of the pristine SS, PVA, and SS/PVA at different ratios are shown in Figure 3. All samples exhibited a porous microstructure associated with the polymer mixture and manufacturing method, with distinct pore configurations influenced by the different polymeric proportions.

The SS sample exhibited a laminar morphology with heterogeneous and undefined pores resembling a honeycomb-like structure. In contrast, the PVA sample showed ribbon-like structures with more organized and compact spaces.

In the SS/PVA scaffolds, the blend ratio significantly affected the microstructure. The 75/25 SS/PVA sample displayed a morphology with open spaces but reduced interconnectivity between pores. As the PVA content increased, the 50/50 SS/PVA scaffold exhibited a more interconnected porous structure with an evident open porosity. However, at the highest PVA proportion (25/75 SS/PVA), the scaffold morphology transitioned to narrower, more compact, and laminar-like porosities, indicating the influence of PVA in forming denser, less open structures. These results highlight the role of PVA content in tailoring the microstructure of the scaffolds, affecting their potential applications where porosity and interconnectivity are critical. This result confirms the physical crosslinking between PVA and SS through hydrogen bonding, as evidenced by FTIR analysis. Similar interactions have been reported in the literature; the structural stability and mechanical characteristics of composite scaffolds are enhanced by the creation of hydrogen bonds between the hydroxyl groups of PVA and amide groups of the protein [20,41].

### 3.3. Performance Analysis

#### 3.3.1. Water Absorption Property and Mass Loss

The water absorption capacity and mass loss, after 24 h of immersion in water for the pristine SS, pristine PVA, and SS/PVA blends with varying ratios, are presented in Figure 4. Pristine SS exhibited a water absorption ratio of 554%, whereas pristine PVA reached a significantly higher ratio of 1970%. For the SS/PVA blends, absorption ratios of 1037%, 1251%, and 1327% were observed for the 75/25, 50/50, and 25/75 ratios, respectively, demonstrating that a higher PVA content correlates with increased absorption capacity.

Pristine PVA exhibited remarkable stability, with negligible mass loss (0.3 ± 0.5%), remaining largely intact. In contrast, pristine SS exhibited a weight loss of 73 ± 3.03%, reflecting significant solubility; however, approximately 27% of the scaffold mass remained intact after immersion. The SS/PVA blends displayed intermediate behavior, with the weight loss decreasing as the PVA content increased by approximately 57%, 52%, and 33% for the 75/25, 50/50, and 25/75 blends, respectively. It is believed that sericin is not the sole component solubilized or released from the scaffold. As sericin dissolves, its interactions with the aqueous medium may destabilize the material, potentially leading to fragmentation and release of small PVA fractions.

The interplay between sericin solubilization and the structural role of PVA highlights the dynamic nature of these materials. Sericin contributes to scaffold bioactivity through its release, whereas PVA enhances structural integrity and ensures functionality in aqueous conditions. The rapid release of sericin within the first 24 h makes these scaffolds effective for applications such as wound healing, in which immediate action is beneficial.

The combination of water absorption capacity and controlled mass loss underscores the potential of these materials for biomedical applications. High absorption facilitates the uptake of wound exudates and promotes the diffusion of bioactive agents while maintaining structural integrity and ensuring reliable performance in moist environments [28,42]. The observed differences in absorption behavior may also result from the pore structures of the scaffolds. Homogeneous pore distribution enhances water retention, whereas heterogeneous pore structures may lead to variations in water content owing to uneven pore sizes [36].

Mass loss was measured to evaluate the degradation behavior of the composite material and its stability in water, which is critical for understanding the performance of scaffolds in moist environments, such as wound beds. Although the scaffolds exhibited notable mass loss during immersion, wound exudates typically involved less liquid exposure than complete submersion. Thus, SS/PVA scaffolds, particularly those with a higher PVA content, strike a balance between absorption and structural stability, making them promising candidates for wound-healing applications.

Interestingly, these results diverge from the existing literature, which generally reports a decrease in absorption capacity as the PVA concentration increases in SS/PVA scaffolds, with SS being the component with the highest absorption ratio [42]. This discrepancy can be attributed to the absence of a physical or chemical crosslinking process in this study. In contrast, most studies utilize crosslinking methods during scaffold fabrication, which significantly affect the absorption behavior.

#### 3.3.2. Protein Release from the Scaffolds

Numerous scientists have emphasized the potential of sericin as a bioactive substance, noting its favorable physicochemical characteristics [43]. In this study, the sericin release test was conducted, as its behavior is associated with future applications in areas such as wound healing, where the amount of the protein compound released influences both its bioactive and cytotoxic responses. Figure 5a,b illustrate the release profiles of silk sericin from scaffolds prepared in different proportions, presented as the amount of sericin released per mL of PBS and as the percentage solubilized relative to the initial sericin content in the material, respectively. Sericin release followed a comparable trend across all samples. During the first 6 h, a gradual release was observed, followed by an accelerated release phase between 6 and 10 h. From 10 to 24 h, sericin was released at a relatively constant rate in the SS/PVA 75/25 and 25/75 samples. However, for the SS and SS/PVA 50/50 samples, a slight decrease in release rate was observed after 10 h.

A significant difference was observed in the total amount of sericin released from the scaffolds. As the proportion of sericin in the samples decreased, the quantity of released protein also decreased substantially. The SS scaffold showed the highest release of sericin, whereas the SS/PVA 25/75 material exhibited the lowest release. This indicated a direct relationship between the sericin content in the scaffold and the quantity released into the medium. Additionally, the SS/PVA 50/50 and 25/75 materials demonstrated a more controlled release over time, which may be attributed to the increased physical interactions, particularly hydrogen bonding, between the sericin and PVA molecules in these blends.

The release of sericin from scaffolds likely occurs via two primary mechanisms: diffusion and material degradation [7,42]. Diffusion dominated during the initial hours (0 to ~10 h), where rapid sericin release was observed, reflecting the mobility of the protein through the porous matrix. On the other hand, degradation-driven release is associated with scaffold solubilization. Based on the experimental results, diffusion appears to be the primary mechanism governing sericin release, as the majority of sericin was released during the initial hours, and no noticeable degradation of the scaffolds was observed throughout the study period [44].

#### 3.3.3. Mechanical Stability: Compression Strength

Figure 6 shows the compressive stress against the percentage of compression deformation of the scaffolds. The curve obtained for PVA showed the highest compressive strength, followed by the scaffold with the lowest sericin content (SS/PVA 25/75). As the proportion of PVA decreased, the compressive strength also decreased.

The results obtained agree with the literature, where it is evident that sericin, which has extremely low mechanical performance, mixed with PVA considerably improves the mechanical properties of the protein [28]. Some investigations have also shown that the presence of PVA in the synthesis of hybrid foams improves the mechanical properties to achieve adequate performance in tissue engineering applications [45].

The stress and strain curves were not linear (see Figure 6b). The behavior of the curve of all scaffolds fits an exponential relationship, which could explain the special viscoelasticity of the evaluated material. The sample with the highest PVA content exhibits the highest compressive stress at each compressive strain level. Therefore, the effect of increasing the PVA content on the stress–strain curve is evident, particularly at high strain rates [46].

## 4. Conclusions

This study evaluated the physicochemical properties and performance of porous scaffolds composed of varying proportions of silk sericin (SS) and poly (vinyl alcohol) (PVA), including at 75/25, 50/50, and 25/75 ratios. FTIR analysis confirmed the successful blending of SS and PVA, with characteristic absorption bands indicating miscibility. The formation of hydrogen bonds between the SS and PVA was evident, contributing to the structural stability and enhanced performance of the scaffolds.

Thermogravimetric analysis (TGA) revealed that the scaffold with a higher PVA content (SS/PVA 25/75) exhibited the highest decomposition temperature, indicating better thermal stability. Morphological analysis showed that higher SS content reduced porosity, while the higher hydrophilicity and water absorption capacity of PVA improved scaffold stability and performance. Scaffolds with a higher PVA content demonstrated better water retention and reduced mass loss, suggesting their suitability for application in moist environments, such as wound healing.

The sericin release profiles indicated a direct correlation between the sericin content and release quantity. The SS scaffold exhibited the highest release, whereas the SS/PVA 25/75 scaffold showed the lowest release. The release process followed two phases: an initial rapid release driven by diffusion and a more controlled release in scaffolds with a higher PVA content, likely due to stronger hydrogen bonding.

In terms of mechanical stability, the PVA scaffold exhibited the highest compressive strength, followed by the SS/PVA 25/75 scaffold. The stress–strain curves displayed exponential behavior, confirming the viscoelastic nature of the scaffolds. A higher PVA content enhanced the compressive strength, suggesting that PVA improves the mechanical stability of scaffolds, making them promising for tissue engineering applications in which structural integrity is essential.

## Figures and Tables

**Figure 1 biomimetics-10-00027-f001:**
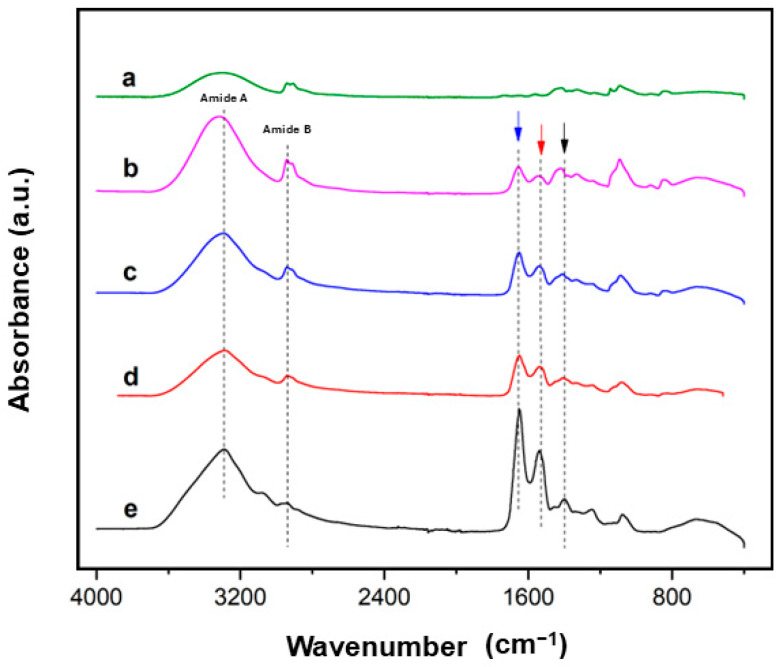
FTIR spectra for (a) pristine PVA, (b) SS/PVA 25/75, (c) SS/PVA 50/50, (d) SS/PVA 75/25, and (e) pristine SS scaffolds.

**Figure 2 biomimetics-10-00027-f002:**
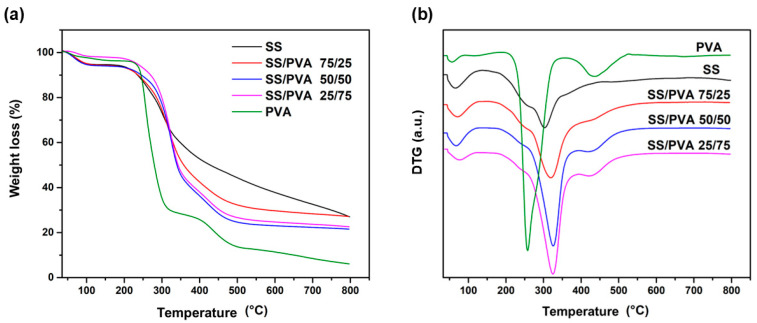
(**a**) TGA and (**b**) DTG curves of pristine SS, PVA, and SS/PVA scaffolds at different proportions of SS/PVA (25/75, 50/50, and 75/25).

**Figure 3 biomimetics-10-00027-f003:**
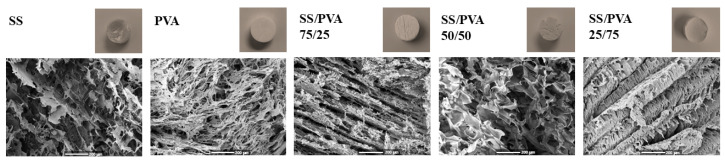
SEM analysis of the cross-sectional images revealing the morphological characteristics of the SS, PVA, and SS/PVA scaffold structures.

**Figure 4 biomimetics-10-00027-f004:**
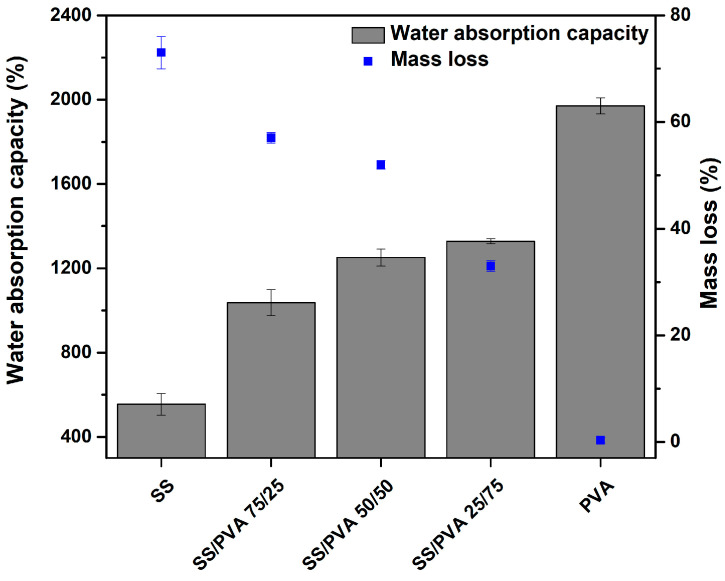
Water absorption after 24 h of immersion in water and mass loss after drying for pristine SS, PVA, and SS/PVA scaffolds at different proportions. Bars represent the standard deviation (*n* = 3).

**Figure 5 biomimetics-10-00027-f005:**
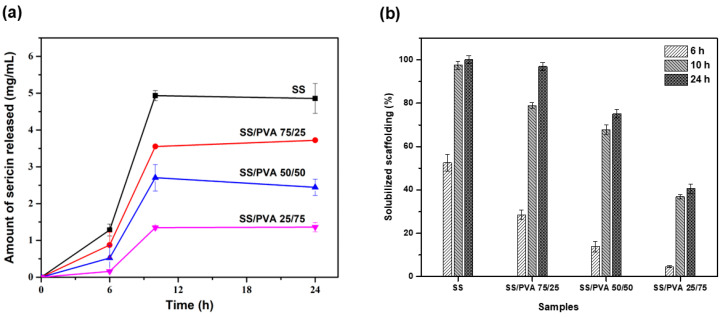
(**a**) Sericin release from SS/PVA scaffolds in PBS and (**b**) percentage of protein released relative to the initial sericin content in SS/PVA scaffolds.

**Figure 6 biomimetics-10-00027-f006:**
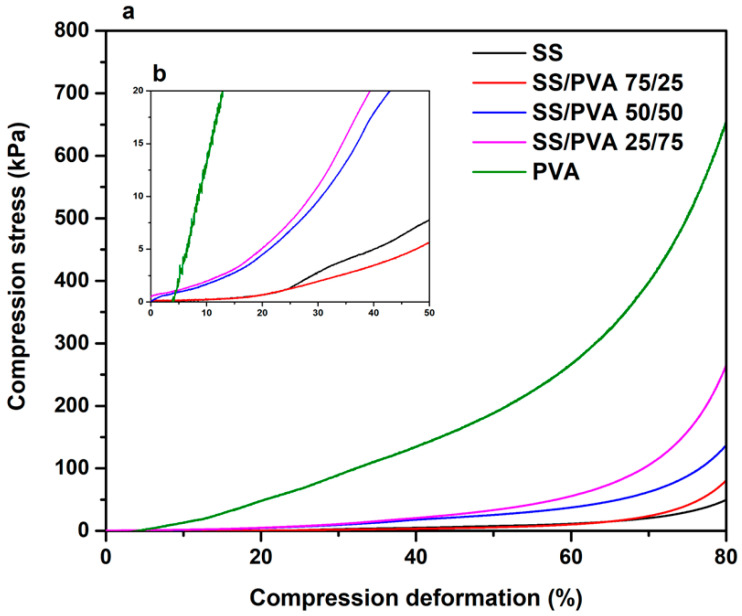
Mechanical integrity of the pristine SS, pristine PVA, and SS/PVA scaffolds in different proportions. (**a**) Full curve and (**b**) magnified initial region.

## Data Availability

The data will be made available upon request.

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
