# Peer review of "Evaluation of Sericin/Polyvinyl Alcohol Mixtures for Developing Porous and Stable Structures"

_biomimetics, 2025, doi:10.3390/biomimetics10010027_

Round 1

Reviewer 1 Report

Comments and Suggestions for Authors

The subject of the manuscript covers study the effects of component ratio on morphology and physical-chemical properties of porous PVA/protein scaffolds fabricated via freeze-drying technique. The research setup is attractive for two reasons. Firstly, PVA is the only polymer capable of forming cryogels with diverse morphology under the conditions proposed by the authors. Secondly, sericin, a waste product of silk production, is used as a filler. This reduces the impact on the environment and complies with the principles of rational use of natural resources. In addition, the selected filler is capable of imparting a number of practically important properties to the materials, namely: antioxidant, antibacterial, anti-inflammatory, etc. However, the presence of highly hydrophobic amino acids in sericin and, as a consequence, the incompatibility of the components makes the use of a hydrophilic polymer matrix very problematic but interesting task. The presence of the functional groups indicated by the authors in sericin does not save the situation, since PVA does not have such for chemical interaction with the filler.

Comments:

1) Lines 69-71 (Materials): Please characterize the sericin, as its physicochemical properties depend on the extraction method and the origin of the silkworm, which can lead to changes in the molecular weight and amino acid concentration. At least according to literary data, for example: Biomed Res Int.; 2016:8175701. doi: 10.1155/2016/8175701

2) Line 145: Please clarify this: “The blank consists of 1 mL of PBS in 10 mL of PBS…”

3) Lines 163 (FTIR analysis): Why is the position of the absorption band of Amide 1 indicated in the text at 1600 cm-1, while in the spectra in Figure 1 (and in the literature) this band appears near 1650?

Lines 164-166: Amide I, Amide II, and Amide III bands are corresponding to the C=O stretching, N-H bending and C-N stretching, respectively. If you agree, please correct the text related to Amide II and III bands.

4) Line 204: What do you mean by side chains of PVA? It is a linear polymer, not branched. The reference you provided (41) is for hydroxyethylcellulose.

5) Lines 222-223: The assertion that the degree of crystallization of sericin is closely related to its thermal stability contradicts what was said above (below as well) about the strong intermolecular interaction between the components, which prevents them from forming their own phase.

6) 3.3.1. Water Absorption Capacity and Mass Loss: The purpose of studying water sorption by immersing samples partially soluble in water is unclear. And it is even more incorrect to designate the obtained data as absorption kinetics. Also, you do not prove the mechanism of absorption in the strict sense of the term. Therefore, I would recommend use of the term "sorption" as well as applying other external conditions (for example, vapor sorption under clearly defined conditions). Also, the paper will not lose anything if you simply remove these data, limiting by studies of protein extraction from the samples.

Lines 295-296: “The SS/PVA 75/25, 50/50, and 25/75 samples exhibited mass losses of approximately 57, 52, and 33%, respectively.” Are these data considering relative amount of sericin in compositions?

7)  Please correct references 19 and 20 in the bibliography.

Author Response

Response to reviewer 1:

The authors sincerely appreciate the reviewers' efforts in evaluating the manuscript, as their feedback has significantly contributed to its improvement. Please find below the responses to each comment.

  1. Lines 69-71 (Materials): Please characterize the sericin, as its physicochemical properties depend on the extraction method and the origin of the silkworm, which can lead to changes in the molecular weight and amino acid concentration. At least according to literary data, for example: Biomed Res Int.; 2016:8175701. doi: 10.1155/2016/8175701

Response: The authors appreciate the reviewer’s valuable comment. To address this point, information regarding the characterization of sericin obtained using the spray-drying method (using Bombyx mori silk cocoons) has been added to the manuscript. This characterization was previously reported in our study (DOI: 10.1134/S0965545X24601011), in which we evaluated the structural and molecular properties of sericin using this specific processing method. The following text has been included in section 2.2. Preparation of Sericin Powder.

 Our previous study [22] detailed the physicochemical characterization of this sericin powder, obtaining a protein with a predominantly amorphous structure and low-molecular-weight fractions.

  1. Line 145: Please clarify this: “The blank consists of 1 mL of PBS in 10 mL of PBS…”

Response: The authors appreciate this observation and added the correct information.

 The blank consisted of 1 mL of PBS in 10 mL of biuret reactive, with absorbance set to 0.0 at 510 nm.

  1. Lines 163 (FTIR analysis): Why is the position of the absorption band of Amide 1 indicated in the text at 1600 cm-1, while in the spectra in Figure 1 (and in the literature) this band appears near 1650? Lines 164-166: Amide I, II, and III bands correspond to C=O stretching, N-H bending, and C-N stretching, respectively. If you agree, please correct the text related to Amide II and III bands

 Response: The authors appreciate this observation and added the correct information.

 The amide I (1650 cm-1, blue arrow), amide II (1515 cm-1, red arrow), and amide III (1240 cm-1, black arrow) bands correspond to C=O stretching, N-H bending, and C-N stretching, respectively [7].

  1. Line 204: What do you mean by side chains of PVA? It is a linear polymer, not branched. The reference you provided (41) is for hydroxyethylcellulose

 Response: The authors appreciate the reviewer's comment and have added the correct term (functional groups). In addition, the reference error has been corrected.

 The second decomposition stage, occurring between 220 and 470 °C, is associated with the elimination of volatile compounds, degradation of amino acid residue side chains in proteins, and decomposition of PVA functional groups [40,41].

  1. Lines 222-223: The assertion that the degree of crystallization of sericin is closely related to its thermal stability contradicts what was said above (below as well) about the strong intermolecular interaction between the components, which prevents them from forming their own phase.

 Response: The authors appreciate the reviewer’s observation and acknowledge the inconsistency in the text. The information has been revised to clarify that the improved thermal stability of the SS/PVA samples is primarily attributed to the strong intermolecular interactions between the components, which restrict polymer chain mobility and increase the energy required for decomposition.

The mention of the degree of crystallization of sericin has been removed to avoid confusion, as the strong interactions between SS and PVA in the blend likely disrupt the formation of distinct crystalline phases. Instead, these interactions promote a physically crosslinked amorphous network, as evidenced by the FTIR analysis. The corrected text now reads:

This enhancement is primarily attributed to the strong physical interactions between the SS and PVA chains, which restrict polymer chain mobility and require more ther-mal energy for decomposition, thereby increasing thermal stability. These findings align with the FTIR analysis, which highlights the physical crosslinking of the poly-mers in the blend and the strong intermolecular affinity between SS and PVA.

  1. 3.1. Water Absorption Capacity and Mass Loss: The purpose of studying water sorption by immersing samples partially soluble in water is unclear. And it is even more incorrect to designate the obtained data as absorption kinetics. Also, you do not prove the mechanism of absorption in the strict sense of the term. Therefore, I would recommend use of the term "sorption" as well as applying other external conditions (for example, vapor sorption under clearly defined conditions). Also, the paper will not lose anything if you simply remove these data, limiting by studies of protein extraction from the samples. Lines 295-296: “The SS/PVA 75/25, 50/50, and 25/75 samples exhibited mass losses of approximately 57, 52, and 33%, respectively.” Are these data considering relative amount of sericin in compositions?

Response: The authors appreciate the reviewer’s comments and have carefully revised the section on water absorption capacity and mass loss to address the concerns raised. The text has been clarified to emphasize that the study focuses on water absorption and mass loss, rather than absorption kinetics, as initially implied. The study now clearly distinguishes between water absorption and mass loss, which occur as a result of the dissolution of sericin and small PVA fractions. Regarding the reviewer’s question about the mass loss data, we confirm that the values presented for the SS/PVA 75/25, 50/50, and 25/75 blends are for total mass loss, without considering the relative amount of sericin in the compositions. This has been clarified in the revised text. The corrected section now reads:

The water absorption capacity and mass loss, after 24 h of immersion in water for the pristine SS, pristine PVA, and SS/PVA blends with varying ratios, are presented in Figure 4. Pristine SS exhibited a water absorption ratio of 554%, whereas pristine PVA reached a significantly higher ratio of 1970%. For the SS/PVA blends, absorption ratios of 1037%, 1251%, and 1327% were observed for the 75/25, 50/50, and 25/75 ratios, respectively, demonstrating that a higher PVA content correlates with increased absorption capacity.

Pristine PVA exhibited remarkable stability, with negligible mass loss (0.3 ± 0.5%), remaining largely intact. In contrast, pristine SS exhibited a weight loss of 73 ± 3.03%, reflecting significant solubility; however, approximately 27% of the scaffold mass remained intact after immersion. The SS/PVA blends displayed intermediate behavior, with the weight loss decreasing as the PVA content increased by approximately 57%, 52%, and 33% for the 75/25, 50/50, and 25/75 blends, respectively. It is believed that sericin is not the sole component solubilized or released from the scaffold. As sericin dissolves, its interactions with the aqueous medium may destabilize the material, potentially leading to fragmentation and release of small PVA fractions.

The interplay between sericin solubilization and the structural role of PVA highlights the dynamic nature of these materials. Sericin contributes to scaffold bioactivity through its release, whereas PVA enhances structural integrity and ensures functionality in aqueous conditions. The rapid release of sericin within the first 24 h makes these scaffolds effective for applications such as wound healing, in which immediate action is beneficial.

The combination of water absorption capacity and controlled mass loss underscores the potential of these materials for biomedical applications. High absorption facilitates the uptake of wound exudates and promotes the diffusion of bioactive agents while maintaining structural integrity and ensuring reliable performance in moist en-vironments [32,47]. The observed differences in the absorption behavior may also result from the pore structures of the scaffolds. Homogeneous pore distribution en-hances water retention, whereas heterogeneous pore structures may lead to variations in water content owing to uneven pore sizes [40].

Although the scaffolds exhibited notable mass loss during immersion, wound exudates typically involved less liquid exposure than complete submersion. Thus, SS/PVA scaffolds, particularly those with higher PVA content, strike a balance between absorption and structural stability, making them promising candidates for wound-healing applications.

  1. Please correct references 19 and 20 in the bibliography.

Response: The authors appreciate this observation and added the correct references.

Reviewer 2 Report

Comments and Suggestions for Authors

This study explored the effects of different SS/polyvinyl alcohol (PVA) ratios on porous scaffolds fabricated via freeze-drying, focusing on the mechanical stability, water absorption, and protein release in phosphate-buffered saline (PBS). This is an interesting and useful investigation. This manuscript can be accepted after some corrections in the following:

1.     The caption of Figure 2 is missing.

2.     In the section of “3.3.1. Water Absorption Capacity and Mass Loss”, the application of water absorption have been explained in the introduction, but the application of MASS LOSS is not shown. Why does the author measure the MASS LOSS of water?

3.     The figure caption of Figure 3a is wrong, which should be corrected.

4.     In the section of “3.3.2. Silk Sericin Release from the Scaffolds”, Why does the author study the Silk Sericin Release from the Scaffolds?

Author Response

Response to Reviewer 2:

The authors sincerely appreciate the reviewers' efforts in evaluating the manuscript, as their feedback has significantly contributed to its improvement. Please find below the responses to each comment.

  1. The caption of Figure 2 is missing.

Response: The authors appreciate the reviewer’s valuable comment. A caption of the figure has been added.

 Figure 2. (a) TGA and (b) DTG curves of pristine SS, PVA, and SS/PVA scaffolds with different proportions of SS/PVA (25/75, 50/50, and 75/25).

  1. In the section of “3.3.1. Water Absorption Capacity and Mass Loss”, the application of water absorption have been explained in the introduction, but the application of MASS LOSS is not shown. Why does the author measure the MASS LOSS of water?

Response: The authors appreciate the reviewer’s comment. The mass loss was measured to evaluate the degradation behavior of the composite material, specifically its stability in aqueous environments. This test is important for understanding how the material might behave under moist conditions, such as those found in wound healing applications. As mentioned in the introduction, the stability of hybrid materials, such as SS/PVA blends, is essential for maintaining their structural integrity and performance during use in humid environments, such as wound beds [18]. The revised manuscript now includes the following clarifications:

Section 3.3.1. Water Absorption Capacity and Mass Loss

Mass loss was measured to evaluate the degradation behavior of the composite material and its stability in water, which is critical for understanding the performance of scaffolds in moist environments, such as wound beds. Although the scaffolds exhibited notable mass loss during immersion, wound exudates typically involved less liquid exposure than complete submersion. Thus, SS/PVA scaffolds, particularly those with a higher PVA content, strike a balance between absorption and structural stability, making them promising candidates for wound-healing applications.

  1. The figure caption of Figure 3a is wrong, which should be corrected.

 Response: The authors reviewed and corroborated this information.

 Figure 3. SEM analysis of cross-sectional images revealing the morphological characteristics of SS, PVA, and SS/PVA scaffold structures..

  1. In the section of “3.3.2. Silk Sericin Release from the Scaffolds”, Why does the author study the Silk Sericin Release from the Scaffolds?

 Response:  The authors appreciate the reviewer’s comment and understand the importance of justifying the study of sericin release from the scaffolds. The release of silk sericin from the scaffolds is crucial because it directly relates to the potential bioactivity and effectiveness of these materials in biomedical applications, particularly in wound healing. Sericin has been recognized for its bioactive properties, including promoting cell adhesion and growth, as well as its antimicrobial and antioxidant effects, which are beneficial for tissue regeneration. By studying the release of sericin, we can better understand the material’s ability to deliver bioactive agents over time, which is essential for applications such as wound dressings, where controlled release of bioactive compounds is key to improving healing outcomes.

 3.3.2. Protein Release from the Scaffolds

Numerous scientists have emphasized the potential of sericin as a bioactive substance, noting its favorable physicochemical characteristics [48]. In this study, the sericin release test was conducted, as it associates its behavior with future applications in areas such as wound healing, where the amount of the protein compound released influences both its bioactive and cytotoxic responses.

Round 2

Reviewer 1 Report

Comments and Suggestions for Authors

The article has been carefully corrected according to all recommendations.

Reviewer 2 Report

Comments and Suggestions for Authors

This manuscript has been much improved, and can be accepted now.